# Common dolphin whistle responses to experimental mid-frequency sonar

Caroline Casey[1,2‡]*, Selene Fregosi[1‡], Julie N. Oswald[3], Vincent M. Janik[3], Fleur Visser[4,5], Brandon Southall[1,2]

1 Southall Environmental Associates, Inc., Aptos, California, United States of America, 2 Institute of Marine Sciences, University of California Santa Cruz, Santa Cruz, California, United States of America, 3 Scottish Oceans Institute, Sea Mammal Research Unit, School of Biology, University of St. Andrews, St. Andrews, United Kingdom, 4 Kelp Marine Research, Hoorn, The Netherlands, 5 Department of Coastal Systems, Royal Netherlands Institute for Sea Research, Den Burg, Texel, The Netherlands

‡ CC and SF are Joint Senior Authors of this work.
* cbcasey@ucsc.edu

**Data Availability Statement:** All detected whistle counts obtained from PAMGuard from the closest buoy at 1-second resolution are available via Dryad

## Abstract

Oceanic delphinids that occur in and around Navy operational areas are regularly exposed to intense military sonar broadcast within the frequency range of their hearing. However, empirically measuring the impact of sonar on the behavior of highly social, free-ranging dolphins is challenging. Additionally, baseline variability or the frequency of vocal state-switching among social oceanic dolphins during undisturbed conditions is lacking, making it difficult to attribute changes in vocal behavior to anthropogenic disturbance. Using a network of drifting acoustic buoys in controlled exposure experiments, we investigated the effects of mid-frequency (3–4 kHz) active sonar (MFAS) on whistle production in short-beaked (*Delphinus delphis delphis*) and long-beaked common dolphins (*Delphinus delphis bairdii*) in southern California. Given the complexity of acoustic behavior exhibited by these group-living animals, we conducted our response analysis over varying temporal windows (10 min– 5 s) to describe both longer-term and instantaneous changes in sound production. We found that common dolphins exhibited acute and pronounced changes in whistle rate in the 5 s following exposure to simulated Navy MFAS. This response was sustained throughout sequential MFAS exposures within experiments simulating operational conditions, suggesting that dolphins may not habituate to this disturbance. These results indicate that common dolphins exhibit brief yet clearly detectable acoustic responses to MFAS. They also highlight how variable temporal analysis windows–tuned to key aspects of baseline vocal behavior as well as experimental parameters related to MFAS exposure–enable the detection of behavioral responses. We suggest future work with oceanic delphinids explore baseline vocal rates a-priori and use information on the rate of change in vocal behavior to inform the analysis time window over which behavioral responses are measured.

data repository (accession number: https://doi.org/10.5061/dryad.d2547d88r).

**Funding:** Funding for this project was provided by the U.S. Navy's Office of Naval Research (Award Numbers N000141713132, N0001418IP-00021, N000141712887, N000141912572). The funders had no role in study design, data collection and analysis, decision to publish, or preparation of the manuscript.

**Competing interests:** The authors have declared that no competing interests exist.

## Introduction

Sound production and reception play a critical role in the lives of cetaceans, aiding in important life-history events, including maintenance of social relationships, coordination of group movement, foraging, and evasion of predators [1]. Consequently, substantial effort has been directed toward describing cetacean acoustic behavior [2,3] and evaluating how it is impacted by human-generated disturbance [4–8]. Many sources of anthropogenic noise pollution (*e.g.*, vessel noise, oil and gas exploration, renewable energy, coastal construction and maintenance, fisheries and aquaculture, and military) can have varying short and long-term impacts on marine mammal behavior and health [4,9–11]. Concentrated research efforts to characterize these impacts have led to the systematic development of acoustic exposure criteria, informing and improving effective management strategies for regulators and industries [for reviews on auditory criteria, see 12]. Such assessments have also highlighted species and noise exposure contexts for which information is sparse or unavailable.

Among cetaceans, oceanic delphinids represent an essential and logistically challenging group to evaluate how anthropogenic noise impacts their vocal behavior. These animals are highly soniferous, abundant, and often extremely gregarious (pods of > 500 individuals are common for some species). Sound production has been demonstrated to play a vital role in the maintenance of social relationships and cohesion among group members [2,3]. Oceanic delphinids are ubiquitous around some U.S. Navy operational areas where mid-frequency active sonar (MFAS; 1–10 kHz) is commonly used for submarine detection in training exercises, resulting in associated large numbers of sonar exposures for these federally protected species. Some of the most powerful MFAS systems (*e.g.*, AN/SQS-53C) emit repeated pings with fundamental frequencies in the 3–4 kHz range, ping lengths of approximately 1–3 s, and nominal source levels as high as 235 dB re 1 μPa at 1 m root-mean-square (RMS) that may be transmitted for several minutes to hours at high duty cycles (more than 1 ping/min) [13]. Aside from the elevated background noise and potential disturbance that may result from these training exercises [*e.g.*, 14], MFAS signals overlap with the frequencies that oceanic dolphins commonly rely on for social sound (whistle) production. Delphinid whistles are narrowband tonal sounds with most acoustic energy concentrated below 20 kHz [15].

While MFAS has been linked to mass stranding events of cetaceans [13,16] and its effect on cetaceans has been experimentally evaluated in a handful of species (for a recent review, see [17]), the impact of sonar on the acoustic behavior of oceanic delphinids has not been systematically explored. This is mainly due to the logistical challenges of applying previously developed methods used in other behavioral response studies of individual animals to large aggregations of dolphins. Much of the prior research on cetacean behavioral responses to noise has capitalized on using suction-cup-attached motion-sensing and acoustic recording tags to characterize responses following controlled exposure to MFAS [*e.g.*, 18–20]. Unfortunately, such tags are challenging to deploy and are easily shed by small dolphins due to the tag size relative to the smaller body surface, high drag due to fast swimming speeds, and frequent physical social contact. Additionally, oceanic delphinids commonly occur in large groups that display remarkable coordination, making the collective vocal behavior of the group perhaps a more appropriate focus of analysis [21]. Opportunistic passive acoustic studies relying on large, cabled hydrophone arrays have been used to quantify changes in vocal activity and thus act as a proxy for the presence or absence of multiple animals before, during, and after exposure to MFAS [22,23]. Such experiments are valuable but require extensive, high-cost moored hydrophone arrays with restricted spatial coverage.

Previous studies on acoustic responses of oceanic dolphins to Navy sonar have observed shifts in frequency-specific components of whistle contours, increasing or decreasing calling

rate, increasing call amplitude, and even mimicry of MFAS elements [14,23–25]. For example, tagged killer whales (*Orcinus orca*) adjust the high-frequency component of their whistles during sonar exposure and increase the number and amplitude of their calls following each ping [24]. False killer whales *(Pseudorca crassidens)* appear to increase their whistle rate and produce more MFAS-like whistles after exposure to simulated sonar [25].

One opportunistic study provided initial insights into the behavioral responses of some social oceanic delphinids to MFAS. Bottlenose dolphins (*Tursiops truncatus*), common dolphins (*Delphinus* sp.), Pacific white-sided dolphins (*Lagenorhynchus obliquidens*), and Risso's dolphins (*Grampus griseus*) incidentally exposed to MFAS showed a cessation of vocalizations, an increase in the intensity of vocalizations, or a combination of both [14]. Of all delphinid species, common dolphins displayed the broadest range of responses, including changing their behavioral state or direction of travel when sonar stopped, increasing the intensity of vocalizations when sonar began, vocalizing very little or not at all during sonar exposure, or a combination of these observations [14]. These results are consistent with a more recent opportunistic evaluation of delphinid responses to an underwater detonation, which showed that whistle rate, complexity, and frequency content varied in response to the explosive event [26]. Depending upon the frequency, intensity, and consistency of these noise exposures, such behavioral changes could result in physiological consequences that impact overall population health [27]. Unfortunately, information on baseline variability and the frequency of vocal state-switching during undisturbed conditions is lacking among free-ranging, social, oceanic delphinids, making it challenging to interpret the responses observed. While opportunistic studies of delphinid acoustic responses to sonar are insightful, a detailed assessment under controlled experimental conditions is needed to understand the extent to which MFAS impacts oceanic delphinids.

Quantifying vocal behavior in these taxa is complicated by the fact that dolphin acoustic behavior is dynamic, variable, and influenced by a myriad of social and environmental factors [28]. Acoustic behavior and how it changes in response to disturbance must be measured across some predetermined time interval. In previous cetacean behavioral response studies using a conventional controlled exposure experimental (CEE) design, this temporal window was often dictated by logistical limitations of the technology being used (*e.g.*, battery power of tags, the feasibility of continuous behavioral observations, etc.), or designed to match the duration of anthropogenic noise source being evaluated. However, averaging vocal behavior over long time windows may result in missing instantaneous or shorter duration responses at the onset of exposure or at scales more biologically meaningful to the individuals exposed. One way to address this is to examine acoustic metrics computed over various time windows to determine if and when we can attribute a change in vocal behavior to a known, controlled disturbance.

For this study, different broad and fine-scale analytical approaches were used to investigate the effects of experimental MFAS on whistle production in short-beaked (*Delphinus delphis delphis*) and long-beaked common dolphins (*Delphinus delphis bairdii*) in southern California. By assessing group-level vocal behavior across different time scales, we aimed to:

1. Describe the variability in baseline vocal behavior of common dolphin aggregations during control conditions.

2. Compare vocal responses detected during a controlled exposure to MFAS across broad and fine temporal scales.

Given the need to establish sampling regimes that can be applied and compared across studies, our objective is to provide an informative framework for assessing the complex

acoustic behavior exhibited by group-living species. We highlight how using different-sized temporal windows–tuned to key aspects of baseline vocal behavior as well as experimental parameters related to MFAS exposure–impacts the detection of behavioral responses.

## Materials and methods

CEEs were conducted with two subspecies of common dolphins. This study was part of a broader effort to quantify group-level responsiveness of oceanic delphinids to military sonar using CEEs around Santa Catalina Island, located off the coast of southern California, USA. Since they regularly occur in mixed groups, we pooled data for the two subspecies to describe baseline vocal data and included subspecies as a potential explanatory variable in our models. The project integrated multiple data streams, including shore-based tracking of dolphin pods, passive acoustics to record vocal activity, and photogrammetry to measure fine-scale behavior [29]. This work was conducted between 2017–2021. We chose our study area because it lies near the Southern California Offshore Range (SCORE)–a tactical training area for the U.S. Navy Pacific Fleet located off the west side of San Clemente Island–where animals regularly encounter the types of signals we used in our experiment.

CEEs comprised three discrete phases: pre-exposure (baseline), exposure using intermittent simulated MFAS signals, and post-exposure. In control trials, the equipment was lowered into the water from the exposure boat but no MFAS signals were broadcast. For details about the experimental source and sound source characteristics–including calculations of received levels–see Durban et al. 2022 [29]. Each experimental phase was 10 min in duration. During exposure phases, MFAS 'pings' of 1.6 s in duration consisting of three tonal and frequency-modulated elements between 3.5–4 kHz were transmitted. Pings were emitted at a broadband source level of 212 dB re 1 μPa RMS every 25 s, which is similar in repetition rate, duty cycle, and the absence of a ramped-up source level (as used in some previous MFAS CEEs) to certain active Navy MFAS systems (*e.g.*, helicopter-dipped sonar systems). Up to 24 total pings were emitted per 10-min exposure phase, provided no permit-mandated shutdowns were implemented for animals within 200 m of the active sound source (this happened in only one CEE). The sound source was positioned relative to focal animal groups using sound propagation modeling to ensure received levels at focal animals were no greater than 140–160 dB re 1 μPa RMS.

For every CEE, subspecies identity (based on differences in genetics, morphology, and pigmentation) was determined using aerial images obtained from drone footage, genetic sequencing from biopsy samples, and visual observation. Additionally, group size was characterized by experienced shore-based observers using binoculars or a binocular scope located at elevated locations (~70 m), enabling a broad overview of the research area (up to 20 km from shore). Shore-based tracking of animals could be conducted for groups up to 7 km from shore. Focal follows included estimating low, best, and high group size, the number of subgroups (defined as all individuals closer to each other than other individuals in the area), the range of inter-individual spacings within subgroups, and distances between subgroups. These observations were taken continuously throughout the experiment at 2-min intervals. For a complete description of these methods, see [29,30].

### Acoustic data collection and processing

**Passive acoustic monitoring.**  Passive acoustic recordings were obtained from each target group of dolphins using three drifting, remote-deployed acoustic recording units. Up to three separate recording units were tactically positioned and recovered from a single small (~6 m) rigid-hull inflatable boat, with the objective of placing one recording unit within 500 m of the

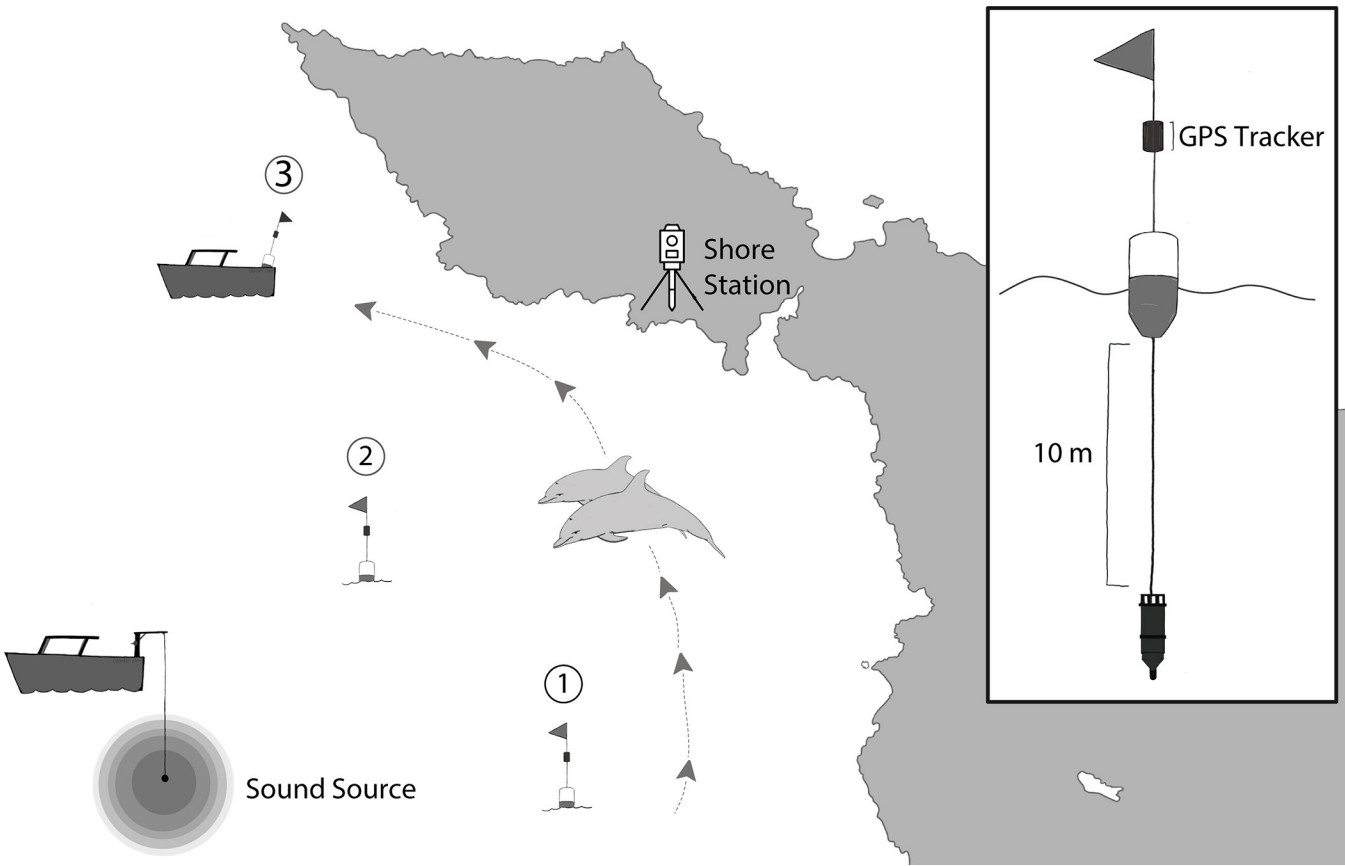

**Fig 1. A schematic representation of the placement of acoustic recorders (see inset for floating acoustic recording unit) relative to the track of the focal group of dolphins.** A single buoy and associated hydrophone were placed within 500 m of the animals during the pre-exposure, exposure, and post-exposure period. The sound source is approximately 1 km from the dolphins at the onset of the exposure period. Note that the source vessel was idling in neutral and was not moving throughout the duration of the exposure period. The dashed line with associated arrows represents the movement path of the focal group. The shore station monitoring the group was positioned on land and is denoted by the theodolite symbol. The map was inspired by images from the NASA Earth Observatory (public domain) and was not drawn to scale (for illustrative purposes only).

predicted trajectory of the dolphins during each CEE phase (Fig 1). Each recording unit consisted of a surface buoy and flag with an underwater recorder. The recorder was either a SoundTrap ST300 (Ocean Instruments NZ, Auckland, New Zealand) or a SNAP Recorder (Loggerhead Instruments, Sarasota, FL, USA). Both recorded via a single omnidirectional calibrated hydrophone (SoundTrap: integrated hydrophone, frequency response 0.02–60 kHz ± 3 dB re 1 µPa, end-to-end sensitivity -178 dB re 1 µPa/V; SNAP: HTI-96-MIN hydrophone, frequency response 1.0–20 kHz ± 3 dB re 1 µPa, end-to-end sensitivity -164 dB re 1 µPa/V,) which was suspended by a shock-mounted cable at a depth of 10 m. All recording units had a Global Positional System (GPS) tracking device (Trace, SPOT LLC, Chantilly, VA, USA) that recorded the location of the instrument once every min (Fig 1). Five-min WAV files were continuously recorded at a 96 kHz sampling rate with a 16-bit resolution (SoundTrap) or a 44.1 kHz sampling rate with a 16-bit resolution (SNAP).

Given the dolphins' frequently unpredictable course, the relative proximity of each hydrophone to the animals was determined post-hoc to evaluate which PAM recording unit was closest to the focal group. The animals' location was known from an associated octocopter drone flight (APO-42, Aerial Imaging Solutions) that utilized a micro 4/3 digital camera (Olympus E-PM2) and 25 mm lens (Olympus M. Zuiko F1.8) mounted to a gimbal. The

octocopter flew at approximately 60 m directly above the dolphins to provide sufficient pixel resolution while decreasing the potential for disturbance [see 29 for details]. The relative distances (in meters) between the focal group (from the drone's GPS) and each recorder (from their flag-mounted GPS units) were estimated for every min of the 30-min experiment using the Haversine formula and linear interpolation in a custom MATLAB script (Mathworks, Natick, MA, USA, Fig 2). Recordings from the buoy closest to the focal group at 1-min intervals were used for all subsequent analyses. Any recordings made when a recorder exceeded 1.6 km from the focal group (even if the recorder was the closest available) were excluded. This threshold was selected based on a previous assessment of detection ranges of playbacks of odontocete whistles (10–20 kHz) by bottom-mounted hydrophones in southern California, which demonstrated a 95% probability of detection of a 135 dB re 1 µPa dolphin whistle at 1.6 km with an SNR of 2.2 dB re 1 µPa [31]. This assessment was supported by the drop-off in whistle amplitude observed in the spectrograms (*post-hoc*) when any buoy surpassed 1.6–1.8 km distance from the focal group (Fig 2). After accounting for this distance cut-off, 9.25 total hours of recordings remained and were used in subsequent analysis.

**Quantifying whistle production and variability.** While common dolphins are known to emit buzzes, echolocation clicks, and whistles, we focused our efforts on characterizing whistle production since they are the critical signal for long-distance communication and play a significant role in group cohesion and coordination [2]. Extraction of whole whistle contours in high background noise with overlapping whistles is exceptionally challenging and results in high rates of missed detections, irrespective of methodology. To create a dataset in which error rates were kept constant across different experimental phases, we used the Whistle and Moan Detector (WMD) module in PAMGuard (v 2.01.05) [32]. The WMD deals with uncertainties by only detecting parts of whistles that clearly stand out above noise using standardized settings across extractions. It is important to note that this often leads to a fragmentation of whistles, with one whistle being split into several independent sections. Thus, whistle detections reported here are not comparable to ones obtained with whole whistle extraction in other studies. However, for our assessment of changes in vocal activity between different experimental phases, it was more important to keep error rates constant to allow for relative comparisons. A qualitative assessment of detector performance within each CEE ensured that variation in whistle detections accurately reflected variation in whistle activity observed in the spectrograms.

The WMD operates on the spectrogram output of the PAMGuard Fast Fourier Transform (FFT) Engine module. We optimized settings for the FFT Engine to provide comparable frequency and temporal resolution of the calculated spectrograms across the two recorders and sampling rates. For the SoundTrap recorders, which had a sampling rate of 96 kHz, the FFT Engine module computed spectrograms with an FFT length of 1024, hop size of 512, and a Hann window. This resulted in a frequency resolution of 93.75 Hz and time resolution of 10.67 ms. For the SNAP recorders, which had a sampling rate of 44.1 kHz, spectrograms were calculated with a Hann window, FFT length 512, and hop size 256, resulting in a frequency resolution of 86.13 Hz and time resolution of 11.61 ms. The WMD was set to detect whistles between 5 kHz and 20 kHz to exclude detection of the tonal sounds from the simulated mid-frequency sonar source (below 5 kHz) and to standardize the upper detection limit across the two sampling rates and avoid any possible edge effects near the Nyquist frequency of the lower sampling rate. The detection threshold was set at 6.0 dB re 1 µPa. Full WMD settings are in the S1 Fig. While the fundamental sonar tonals were excluded by the 5 kHz high-pass cut-off for detections, the high source level of the simulated MFAS resulted in the presence of harmonics in some of the recordings. All harmonics were manually annotated in PAMGuard Viewer using the Spectrogram Annotation module for later removal.

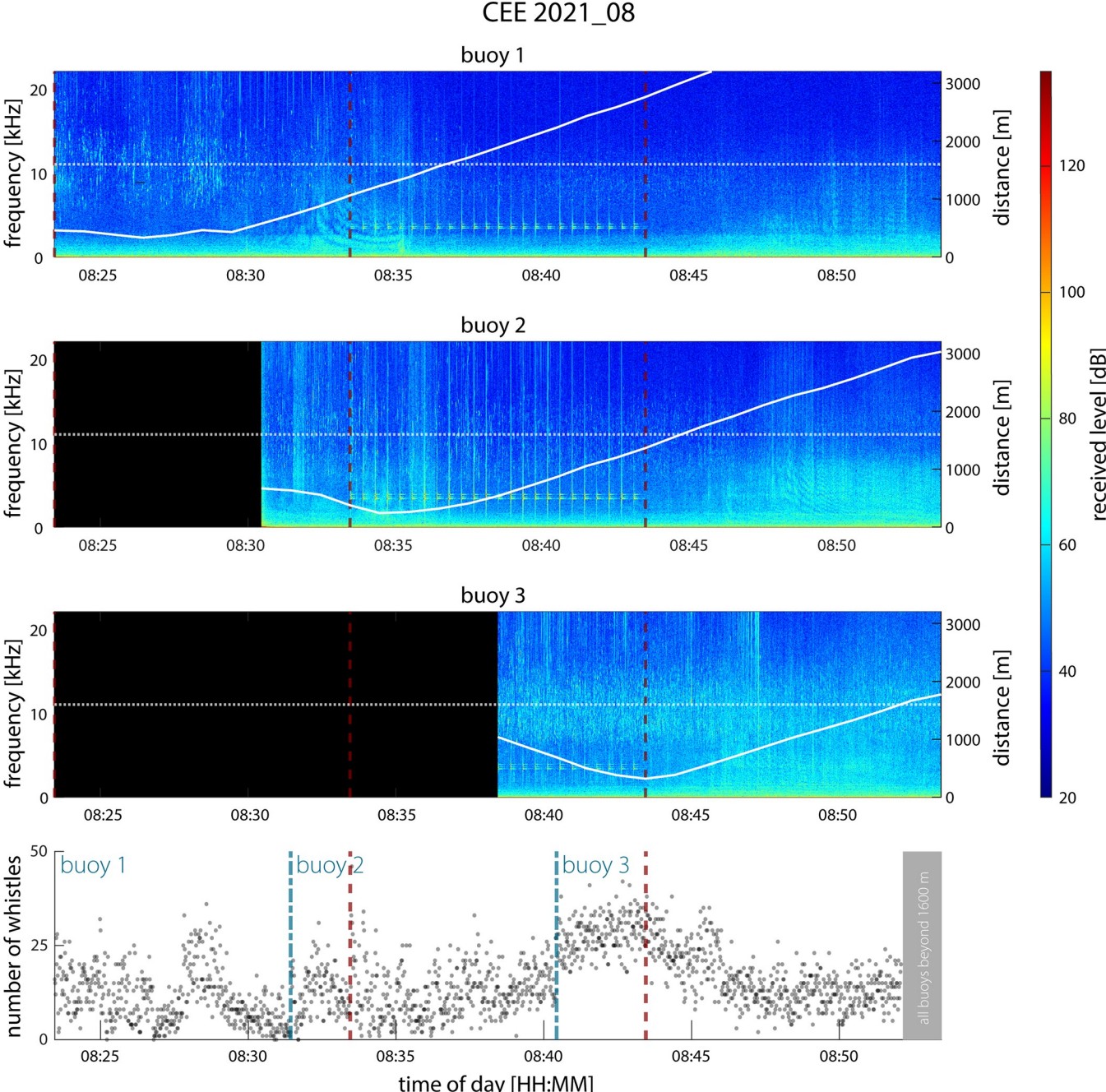

**Fig 2. Spectrograms of each of the three recorders strategically placed on the track line of a moving group of dolphins and associated whistle detections from the buoy closest to the animals at each minute.** The solid white line represents the distance between the recorder and the drone centered over the focal group (units on the right y-axis), the red dashed vertical lines denote the experimental period, and the horizontal white dashed line marks the 1.6-km threshold. Estimates of the relative distance between the focal group and each recorder were assessed every min of the 30-min experiment. The bottom panel shows the number of whistles detected on the closest recorder using the PAMGuard Whistle and Moan Detector. The blue dashed lines and associated blue numbers indicate times when the closest buoy switched, and which buoy was closest. The grey area denotes where whistle detections were excluded due to the recorder distance exceeding 1.6 km.

Detected whistles were automatically exported from the PAMGuard detection database using PAMGuard MATLAB tools (https://github.com/PAMGuard/PAMGuardMatlab).Annotated MFAS harmonics were removed using R package 'PAMmisc'in R version 4.3.1 [33,34]. Whistles were quantified at 1-s resolution; because whistles were often longer than 1 s in duration, the total number of whistles starting within a 1-s bin was counted, providing a metric for whistle activity as whistles detections per second. For brevity, this is referred to as whistle count throughout the remainder of this manuscript.

## Characterization of baseline vocal behavior

**Overall whistle count.** We calculated the mean and median whistle count per second for the entire 30-min experiment for each control CEE to assess common dolphin vocal behavior under control conditions. These data were evaluated with reference to group size estimates collected by experienced shore-based observers.

**Changepoint analysis.** We applied a changepoint analytical approach to the control CEE data collected for both common dolphin subspecies to describe the natural variability in vocal behavior during control conditions. Change point detection is used to pinpoint times when the probability distribution of a time series changes (*i.e.*, vocal state changes). The aim is to identify times when either the mean or variance deviates from the expected trends in the dataset and estimate the number and position of all changepoints. Effectively, this approach detects points in time when a significant change in whistle count occurs. First, a 5-s smoothing window was applied to the raw 1-s whistle count data. Then, changepoints in both mean whistle count and whistle count variance were detected using the 'changepoint' package in R version 2.2.4 [34,35]. The "BinSeg" (Binary Segmentation) algorithm was used. This provided the number and locations of all state changes in the mean and the variance of whistle count over the 30-min sampling period.

**Assessing the impact of disturbance on vocal behavior.** Using broad and fine-scale time windows, we employed a hierarchical approach to characterizing the types of vocal responses that might be detected during controlled exposure to MFAS (Fig 3). All statistical analyses were carried out using R version 4.3.1 [34].

**Difference in changepoints by period across CEEs.** We conducted a changepoint analysis on all CEEs (both controls and MFAS) to evaluate whether common dolphins change the frequency of vocal state switching as a result of exposure to MFAS. We used the same general method described above but quantified the number of changepoints in the 10-min pre-exposure and 10-min exposure periods separately. Changepoints were detected for both the mean and variance of the whistle count data. The difference in the number of changepoints between the two periods was calculated, and an unpaired t-test was used to evaluate any significant differences between controls and MFAS experiments.

**Characterize the impact of MFAS exposure on whistle count: 10-min time scale.** To identify potential broad-scale changes in whistle count in response to simulated MFAS exposure, we pooled and analyzed all CEEs (both controls and MFAS exposures) using a Generalized Linear Mixed Model (GLMM) approach, implemented using R package 'glmmTMB'[36]. We modeled the absolute difference in median whistle count between the 10-min pre-exposure period and the 10-min exposure period (*periodDiff*) as a function of CEE type (*ceeType*; either control or MFAS), a random identity variable (*ceeNum*), dolphin subspecies (*subSpecies*), the best estimate of total group size from the shore-based observers (*groupSize*), and the mean distance between the focal group and the closest buoy for the entire CEE (*buoyDistance*). Using the absolute value for difference in median whistle count enabled us to explore the magnitude of a potential response. We modeled the relationship using a negative binomial distribution, which fit the count-type data after the transformation. Our full model was:

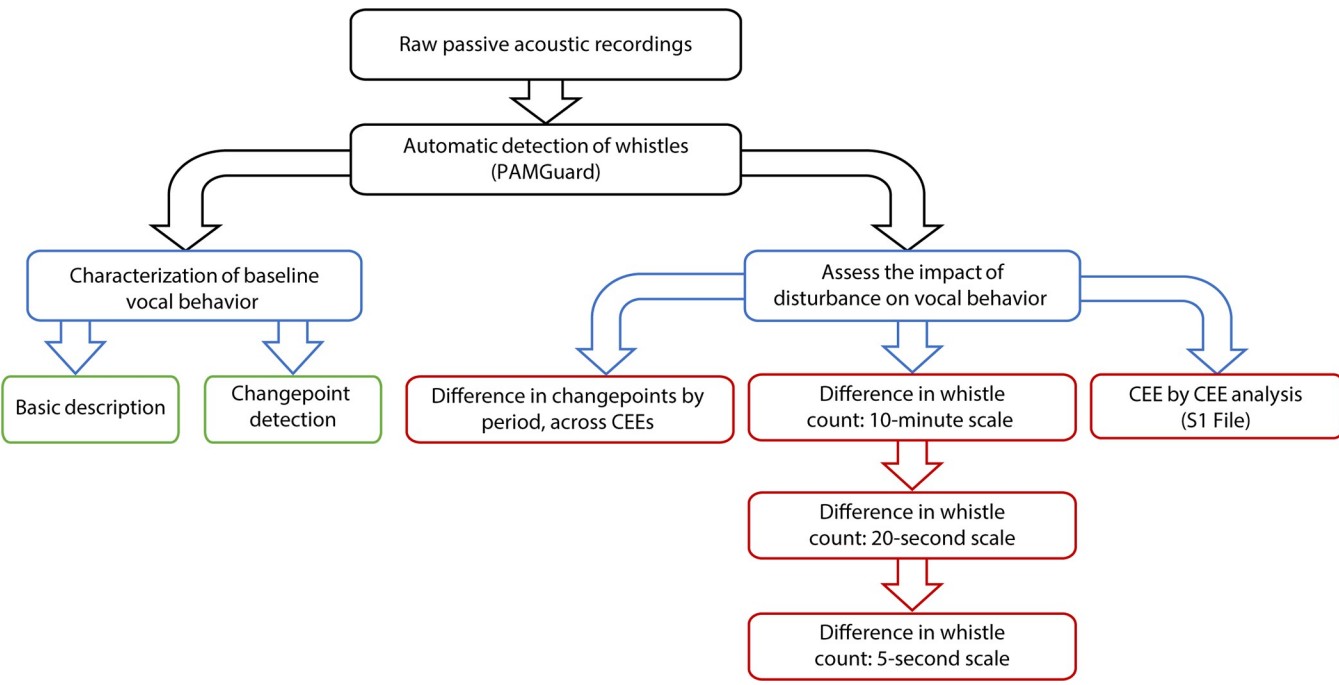

**Fig 3. Flowchart of methods implemented to assess changes in common dolphin vocal behavior during controlled exposure to MFAS.** Methods include pre-processing of acoustic data and baseline vocal behavior analysis as well as a hierarchical assessment of disturbance on vocal behavior at three temporal scales.

$$periodDiff \sim ceeType + ceeNum + subSspecies + groupSize + buoyDist$$

We used backward elimination, ΔAIC, and analysis of variance (ANOVA) to select the best model.

**Characterize the impact of MFAS exposure on whistle count: 20-s time scale.** To characterize more instantaneous changes in whistle production in response to MFAS exposure, we compared dolphin whistle detections in the 20 s before and 20 s after each ping (n = 24 1-s pings per 10-min experimental period, ~25 s between each ping) for both MFAS experiments and controls. We selected this time window to capture sustained variation in whistling behavior within a single ping cycle without overlap between cycles. Differences between these two sequential time bins were calculated by subtracting the mean whistle count for the first bin from the mean whistle count of the second bin (Fig 4B). The first ping started at time 0, the second at 25 s, and so on. Because no actual pings were present in the control experiments, we calculated the change in whistle count surrounding time points placed at the same time as when actual pings would have occurred during an MFAS CEE.

Like the analysis at the 10-min time scale, we used a GLMM approach (using R package 'glmmTMB') to identify potential significant differences immediately following pings (*ping-Change*) in MFAS experiments compared to controls where no pings were present. In addition to the previous fixed effects included at the 10-min scale (*ceeType, ceeNum, subSpecies, buoy-Dis, and groupSize*), we also included median whistle count per second for the entire experimental period to account for the varied baseline whistling activity across CEEs (*medWhist*).

$$pingChange \sim ceeType + ceeNum + subspecies + groupSize + buoyDist + medWhist$$

Binned ping change data were generally normally distributed but were zero-inflated, so we used a Gaussian distribution for the primary model and additionally modeled the zero inflation as a function of median whistle count (*medWhist*). We used backward elimination, ΔAIC, and analysis of variance (ANOVA) to select the best model.

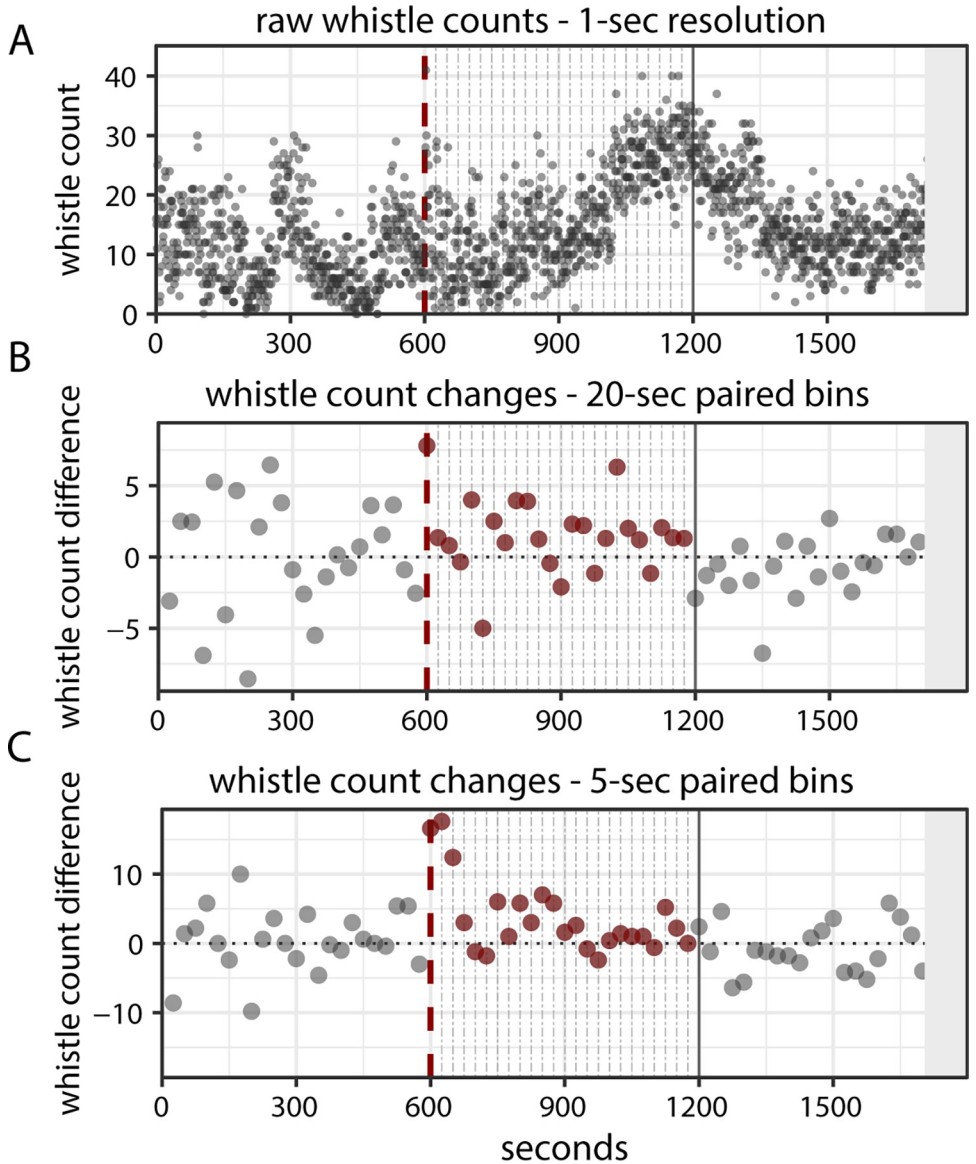

**Fig 4.** Example plots of (A) raw whistle detections over time, (B) changes in whistle count between 20-s duration sequential bins, and (C) changes in whistle count between 5-s duration sequential bins for the pre-exposure, exposure, and post-exposure period within one CEE. The dashed vertical red line indicates the onset of exposure, and the sequential gray dashed lines represent each ping within the exposure period. Post-exposure periods were not included in the modeling analysis but are presented here for reference. The grey shaded area denotes where whistle detections were excluded due to the recorder distance exceeding 1.6 km.

**Characterize the impact of MFAS exposure on whistle count: 5-s time scale.** We repeated the analysis conducted above but over a shorter 5-s time window to investigate potential instantaneous changes immediately following pings (Fig 4C).

Like the above analysis, we used a GLMM approach to identify potential significant differences in whistle count changes in the 5 s following compared to the 5 s before each ping (*ping-Change*) for MFAS experiments compared to controls. We used the same fixed effects implemented at the 20-s scale (c*eeType, ceeNum, [subSpecies, buoyDis, and groupSize, medWhist)* but also included an autocorrelation structure to this analysis *AR(1)* to account for apparent temporal lag effects in exploratory plots.

$$pingChange \sim ceeType + ceeNum + subSspecies + groupSize + buoyDist + medWhist + AR$$
(1)

Like the 20-s scale, binned ping change data were normally distributed and zero-inflated; a Gaussian distribution was used for the conditional model and zero-inflation was modeled as a function of median whistle count (*medWhist*). We used backward elimination, ΔAIC, and ANOVA to select the final model.

**CEE-by-CEE analysis.** We assessed each playback individually at each time scale to better contextualize the severity and persistence of responses and whether dolphins increased or decreased their whistle behavior following sonar exposure. The methods and results for the CEE-by-CEE analysis can be found in the supplementary materials (S1 File).

## Results

### Characterization of baseline vocal behavior

**Basic description.** This analysis includes nine control experiments, each conducted on separate days. Four control experiments were conducted with short-beaked common dolphins, and five were conducted with long-beaked common dolphins (Table 1). This resulted in 270 mins of baseline acoustic data for both subspecies (pooled). The average group size across subspecies was 190 individuals (45–300 animals). The dispersion of animals varied considerably within and between control experiments, including small to large groups (55–300 individuals) in tight to lose organization, joint (in a single group with no subgroups), or spread out over several subgroups (range: 2–6) at tens to several hundreds of meters apart (range: 10–800 m).

Whistles were successfully detected across all control experimental deployments. Mean (SD) and Median (IQR) whistle count per second varied between control CEEs (Table 1). In our assessment of the control experiments, we found the total number of whistles varied between 0.3–4.6 whistles/s. Note that the inherent uncertainty error in our group size assessment for large groups did not allow us to calculate accurate whistle rates/individual (see Table 1 for group size estimates and whistle detections).

CEE ID denotes the year and the CEE number. Subspecies abbreviations are Db for *D. d. bairdii* and Dd for *D. d. delphis*, and the estimated group size is taken from shore observations. Received sound levels (RLs) are reported by Durban et al. 2022 [29]. The median and interquartile range (IQR, 25-75th percentiles) are given for the raw whistle detections per second of the entire 30-min CEE. Changepoints were calculated for both the mean whistle count and the variance in whistle count, separately for the pre-exposure and exposure periods. Note that "exposures" in controls were quiet periods compared to sound exposure in MFAS trials.

**Baseline changepoint analysis.** The baseline changepoint analysis revealed that mean detected whistle counts over the 30-min sampling periods changed once every min, and variance in detected whistle counts changed once every 3.5 min.

### Assessing the impact of disturbance on whistle behavior

A total of 10 MFAS CEEs were conducted–seven of which included long-beaked common dolphins and three of which included short-beaked common dolphins. The calculated average received level across all experiments was 151 dB re 1 μPa RMS (range 142–159 dB re 1 μPa RMS, Table 1). The average group size for MFAS CEEs was 173 individuals (range 10–300, Table 1). For the changepoint analysis, all 10 MFAS experiments were included. However, for the assessment of changes in whistle behavior across different time scales, CEE 2021_11 (conducted with long-beaked common dolphins) had to be excluded because the limited number of detected whistles could not be successfully modeled. Consequently, the modeling results include nine controls and nine MFAS experiments.

**Table 1. Summary of each controlled exposure experiment, including controls (no sound emitted) and MFAS (playback of mid-frequency active sonar).**

| CEE ID | Subspecies | Estimated group size | Type | RL (max) | RL (range) | Median [IQR] whistles per second | Changepoints (mean, pre-exposure) | Changepoints (mean, exposure) | Changepoints (variance, pre-exposure) | Changepoints (ariance, exposure) |
|---|---|---|---|---|---|---|---|---|---|---|
| 2019_01 | Db | 260 | MFAS | 147 dB re 1 μPa RMS | 140–147 dB re 1 μPa RMS | 6.96 [6.39] | 26 | 39 | 5 | 1 |
| 2019_02 | Dd | 350 | control | n/a | n/a | 4.80 [5.19] | 37 | 29 | 0 | 5 |
| 2019_04 | Db | 200 | control | n/a | n/a | 0.25 [0.77] | 2 | 0 | 4 | 2 |
| 2019_06 | Db | 45 | control | n/a | n/a | 0.32 [1.38] | 5 | 0 | 4 | 9 |
| 2019_07 | Db | 300 | MFAS | 154 dB re 1 μPa RMS | 150–154 dB re 1 μPa RMS | 3.11 [3.81] | 36 | 20 | 3 | 3 |
| 2019_08 | Db | 250 | MFAS | 142 dB re 1 μPa RMS | 131–142 dB re 1 μPa RMS | 1.50 [3.91] | 30 | 12 | 4 | 4 |
| 2019_09 | Dd | 250 | control | n/a | n/a | 4.21 [5.38] | 46 | 44 | 2 | 4 |
| 2019_10 | Dd | 30 | MFAS | 149 dB re 1 μPa RMS | 146–149 dB re 1 μPa RMS | 0.091 [0.38] | 0 | 0 | 3 | 7 |
| 2021_01 | Db | 150 | control | n/a | n/a | 3.14 [4.47] | 3 | 51 | 1 | 3 |
| 2021_02 | Db | 200 | control | n/a | n/a | 4.68 [3.89] | 18 | 29 | 0 | 4 |
| 2021_03 | Dd | 150 | control | n/a | n/a | 0.99 [2.01] | 11 | 0 | 2 | 0 |
| 2021_04 | Db | 150 | control | n/a | n/a | 4.66 [4.20] | 27 | 37 | 4 | 2 |
| 2021_05 | Dd | 250 | control | n/a | n/a | 0.36 [1.87] | 0 | 5 | 8 | 4 |
| 2021_08 | Db | 30 | MFAS | 153 dB re 1 μPa RMS | 145–153 dB re 1 μPa RMS | 14.13 [7.85] | 50 | 49 | 2 | 4 |
| 2021_09 | Db | 200 | MFAS | 157 dB re 1 μPa RMS | 152–157 dB re 1 μPa RMS | 1.17 [2.17] | 3 | 6 | 2 | 2 |
| 2021_10 | Db | 300 | MFAS | 159 dB re 1 μPa RMS | 150–159 dB re 1 μPa RMS | 15.34 [9.16] | 42 | 61 | 3 | 2 |
| 2021_11 | Db | 10 | MFAS | 153 dB re 1 μPa RMS | 150–153 dB re 1 μPa RMS | 0.0029 [0.063] | 0 | 0 | 0 | 6 |
| 2021_12 | Dd | 150 | MFAS | 152 dB re 1 μPa RMS | 149–152 dB re 1 μPa RMS | 2.85 [3.07] | 7 | 24 | 4 | 6 |
| 2021_13 | Dd | 200 | MFAS | 147 dB re 1 μPa RMS | 139–147 dB re 1 μPa RMS | 2.21 [4.73] | 2 | 2 | 9 | 4 |

**Number of changepoints in pre-exposure vs exposure.** Changepoint analysis was run for all control (9) and MFAS (10) experiments. The number of changepoints detected in both the mean and variance of whistle count during the pre-exposure and exposure periods is presented in Table 1 for both controls and MFAS experiments. There was no significant increase or decrease in the mean and variance of detected whistle counts following MFAS exposure when compared to the natural variance present during control conditions. The difference (Δ) in change points in variance of whistle detections between the pre-exposure and exposure period did not differ significantly between controls and MFAS CEEs (controls: M = 3.1, SD = 1.4; MFAS: M = 2.5, SD = 2.3; t(17) = 0.69, p = 0.5). The same was true when comparing the mean whistle count between both experimental types (controls: M = 11.3, SD = 14.2; MFAS, M = 8.7, SD = 8.5; t(17) = 0.5, p = 0.63).

**Impact of MFAS exposure on whistle count: 10-min time scale.** Whistle detections did not change between the pre-exposure and exposure period during MFAS experiments at the 10-min time scale. The preferred model was the simplest model with the absolute value of the change in median whistle count as a function of only CEE type (either MFAS or control). There was no significant effect of CEE type on the change in median whistle detections (negative binomial GLMM, n = 18, p = 0.8). The full model (ΔAIC 5.4) indicated that no proposed predictor variables (CEE type, CEE number, subspecies, buoy distance, or group size) had a significant effect on the change in median whistle count between the pre-exposure and exposure periods for both MFAS and control CEEs (negative binomial GLMM, n = 19: P > 0.05 for all variables, Table 2).

**Characterize the impact of MFAS exposure on whistle count: 20-s time scale.** Whistle detections did not differ significantly over the 20-s time windows surrounding each ping. The preferred model at the 20-s scale included only predictor variables for CEE type and median whistle count (Table 2). The results of this model showed that CEE type did not have a significant effect on changes in whistle count in the 20 s after each ping (GLMM, n = 18, slope = 0.47, SE = 0.27, P > 0.05, Table 2) but that the baseline median whistle count for that

**Table 2. Overview of GLMMs used at three-time scales– 10-mins, 20-s, and 5-s.**

| Conditional model | Zero-inflation model | Distribution | ΔAIC | Degrees of freedom | Dispersion ($\sigma^2$) |
|---|---|---|---|---|---|
| 10-min scale | | | | | |
| abs(*periodDiff*) ~ *ceeType* | n/a | nbinom2 | 0 | 3 | 5.44 |
| abs(*periodDiff*) ~ *ceeType* + (1 \| *ceeNum*) | n/a | nbinom2 | 2 | 4 | 1.04 |
| abs(*periodDiff*) ~ *ceeType* + (1 \| *ceeNum*) + *subSpecies* + *buoyDist* + *groupSize* | n/a | nbinom2 | 5.4 | 7 | 1.48 |
| 20-s scale | | | | | |
| *pingDiff* ~ *ceeType* + *medWhist* | ~*medWhist* | gaussian | 0 | 6 | 7.1 |
| *pingDiff* ~ *ceeType* | ~*medWhist* | gaussian | 5.1 | 5 | 7.23 |
| *pingDiff* ~ *ceeType* + *subSpecies* + *groupSize* + *buoyDist* + *medWhist* | ~*medWhist* | gaussian | 5.3 | 9 | 7.09 |
| *pingDiff* ~ *ceeType* + (1 \| *ceeNum*) | ~*medWhist* | gaussian | 7.1 | 6 | 7.23 |
| *pingDiff* ~ *ceeType* | none | gaussian | 28.3 | 3 | 6.71 |
| 5-s scale | | | | | |
| *pingDiff* ~ *ceeType* + *groupSize* + *medWhist* + ar1(*times* + 0 \| *ceeNum*) | ~*medWhist* | gaussian | 0 | 9 | 10.1 |
| *pingDiff* ~ *ceeType* + (1 \| *ceeNum*) + *subSpecies* + *groupSize* + *buoyDist* + *medWhist* + ar1(*times* + 0 \| *ceeNum*) | ~*medWhist* | gaussian | 2.6 | 12 | 10.1 |
| *pingDiff* ~ *ceeType* + ar1(*times* + 0 \| *ceeNum*) | ~*medWhist* | gaussian | 13.8 | 7 | 9.83 |
| *pingDiff* ~ *ceeType* + (1 \| *ceeNum*) + ar1(*times* + 0 \| *ceeNum*) | ~*medWhist* | gaussian | 15.8 | 8 | 9.83 |
| *pingDiff* ~ *ceeType* | none | gaussian | 207.8 | 3 | 10.9 |

Model formulas (conditional and zero-inflation if included) are listed for each time scale of analysis, and within each time scale are given in ascending order of ΔAIC relative to the best model (ΔAIC = 0).

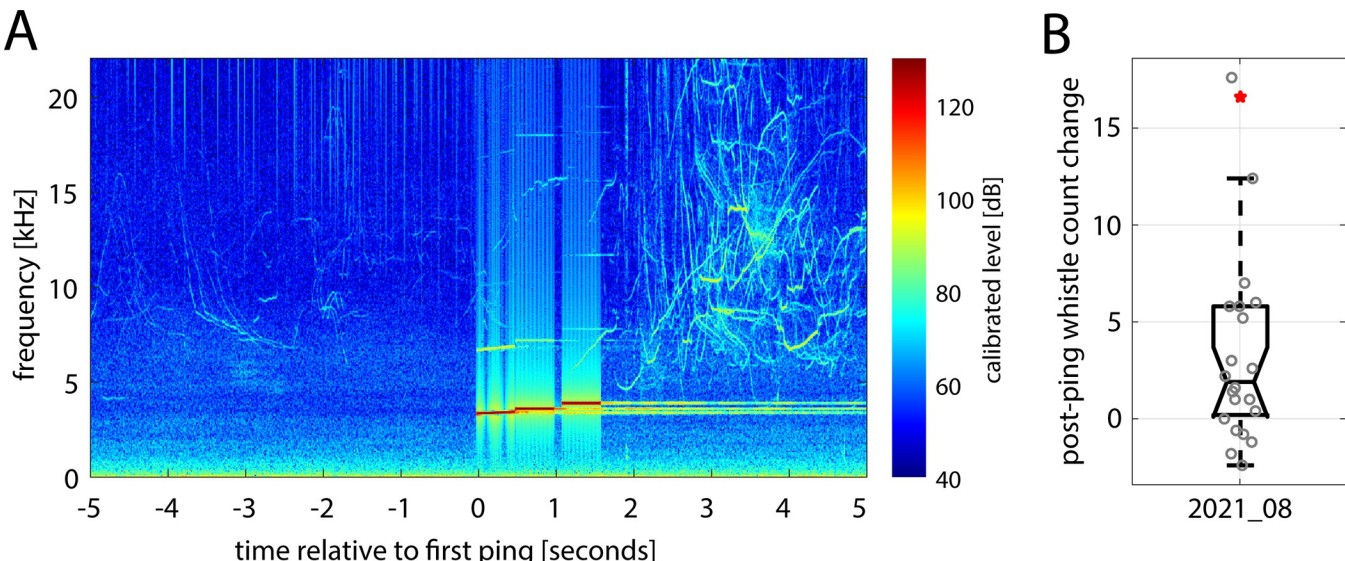

**Fig 5. Acoustic response of common dolphins to the first ping of experimental sonar.** (A) Spectrogram example of 5 s before and 5 s after the first ping for MFAS CEE 2021_08 illustrates the significant increase in whistle count immediately following the cessation of the ping. The focal group was comprised of approximately 30 long-beaked common dolphins. The MFAS signal can be seen between 3 and 4 kHz. (B) Boxplot of the change in whistle count from the 5 s before to the 5 s following each of the 24 pings for CEE 2021_08. Boxplot shows median, 25th, and 75th percentiles, with raw whistle count changes as open gray circles. The change following the first ping is shown as a red star.

experimental period was a significant predictor for the change in whistle count following a ping or control treatment (GLMM, n = 19, slope = 0.76, SE = 0.028, p = 0.0075).

**Characterize the impact of MFAS exposure on whistle count: 5-s time scale.** The preferred model at the 5-s scale included the temporal autocorrelation structure and three explanatory variables, CEE type, group size, and median whistle count, all of which had a significant effect on the change in whistle count in the 5 s immediately following a ping compared to the 5 s immediately before a ping. When accounting for all other variables, the magnitude of the change in whistle count in the 5 s following an actual MFAS ping was 1.4 times greater than any change in whistle count following control treatments (GLMM, n = 18, slope = 1.43, SE = 0.47, p = 0.002, Table 2). The results of the CEE by CEE analysis (S1 File) showed that in five of the nine MFAS experiments, detected whistle counts were elevated in the 5 s after each ping for the entire 10-min exposure period (plots of all raw whistles are provided in S2 Fig). The effect was particularly pronounced (outside the 75th percentile; S3 Fig) in the first ping of six of the MFAS CEEs; whistle activity increased in the 5 s following the first MFAS ping up to 15 times the whistle count in the 5 s before the first ping (mean of all MFAS CEEs 3.9, SD 5.2), compared to increases of only up to 1.4 times (mean 0.46, SD 0.88) at the start of control treatments (Figs 5 and S3). Additionally, group size and median whistle count for the exposure period were significant predictor variables. Larger groups showed more extensive changes in whistle count following pings and control treatments (GLMM, n = 18, slope = 0.007, SE = 0.003, p = 0.008, Table 2), and when the median background whistle count was higher, so too were the changes following pings and control treatments (GLMM, n = 18, slope = 0.25, SE = 0.048, p = < 0.005, Table 2).

## Discussion

Multiple factors–including rapid changes in behavioral state throughout the experiment and variation in group size and composition–make it difficult to assess whether changes in vocal

behavior are due to disturbance or natural variability. We assessed vocal behavior during control conditions to understand typical acoustic variation among common dolphins. We found that dolphins exhibited natural vocal state changes (identified by the changepoint analysis) in whistle production approximately once every min. This rapid acoustic state switching informed our analytical approach, which utilized a range of temporal windows to test for changes in whistle count (5 s– 10 min) before and after MFAS exposure. Across the longer time windows selected; we did not detect a shift in whistle behavior attributable to MFAS. However, CEE type significantly affected the change in whistle count in the 5 s following a ping compared to the 5 s immediately before a ping.

The initial selection of the time period over which behavior was assessed in response to Navy sonar (10 min) for this project was influenced by multiple factors, including the flight endurance of the drone used for calculating animal distance to our recording buoys, our ability to consistently track large groups of fast-moving dolphins, and other previous BRS studies using MFAS [*e.g.*, 18,37]. Many of the aforementioned constraints are imposed by the logistics of fieldwork. An informed approach to identifying behavioral responses to anthropogenic sources also requires prior knowledge of the timing of behavioral state switching in the study species. For example, previous work with blue whales (*Balaenoptera musculus*) evaluated several behavioral metrics (*e.g.*, maximum depth, dive time, ascent/descent rate) in responses to simulated MFAS exposure over a 30-min time window [37]. This exposure duration (which included a 30-min pre-exposure period) was adequate to capture a behavioral change given the typical duration of their dive cycles (5–8 mins) [38]. In contrast, beaked whales exhibit incredibly long, deep foraging dives that often last over an hour, followed by long periods of recovery [*e.g.*, 39,40]. Consequently, studies focused on direct measurements of behavioral response by Cuvier's beaked whales to MFAS extended their pre-exposure baseline period up to 9.3 hours and evaluated their response to sonar for up to 1.7 hours after the exposure period [41].

While the 10-min exposure period seemed appropriate given the fast-paced lifestyle of common dolphins, neither the aggregate model nor the individual assessment of each CEE detected a change in acoustic behavior that could be ascribed to sonar exposure at this time scale. Even when we explored variation in whistle production during the 20-s surrounding each ping, the experimental period was not a significant predictor of changes in whistle count in either the MFAS or control experiments, as vocal state switching often occurs within a 40-s time window under baseline conditions. While the impacts of sonar may be evident over the 10-min exposure period for other behavioral metrics (*i.e.*, changes in behavioral state, group composition, diving behavior), our analysis reveals that changes in acoustic behavior are limited to an extremely narrow time window in these two subspecies.

It was only at the 5-s time scale surrounding each ping that we observed dolphins exhibiting an acute acoustic response, which included a rapid increase in whistle production relative to the 5 s immediately before sound exposure. On average, dolphins increased their whistle count four times the average in the 5 s preceding the first ping of the exposure. In one MFAS experiment, dolphins increased their whistle production 15 times compared to the whistle count in the 5 s immediately before the first ping (S3 Fig). This elevated vocal response following the first ping of the exposure was seen in six of the nine MFAS exposures where whistles were present.

Elevations in whistle detections did not occur during the sonar transmission itself–which lasted for 1.6 s. Rather, the increased vocal production occurred once the signal had been transmitted, often abating within ~ 10 s. The lack of whistle production during sonar transmission may be a tactic for reducing acoustic interference and masking, which has been shown to impact the detection, discrimination, and localization of relevant signals [42]. If the signal is predictable (as in our experiment), then animals should be able to adjust the timing of sound production to limit communication to periods in which noise is reduced [*e.g.*, 42–45]. The

ability of dolphins to learn the timing of intermittent noise has previously been demonstrated by Finneran et al., 2023 [42], who showed that individuals can modify their hearing sensitivity before the onset of predictably timed impulses, presumably to mitigate adverse auditory impacts. Surprisingly, little is known about their capacity to alter the timing of vocal production in response to interfering signals under baseline conditions.

The sudden increase in vocal behavior following the first ping could be an example of the amplification of the behavior of group members through recruitment or reinforcement (*i.e.*, positive feedback) [46]. In this scenario, one dolphin may whistle in response to a surprising, salient stimuli, and others may follow suit. As this recruitment response continues, the number of dolphins producing whistles will increase further, and information is spread rapidly throughout the group [47]. In such cases, a few key individuals could catalyze the collective behavior of the rest of the group. While it would be difficult to evaluate this process using acoustic data alone, concurrent video data collected from the associated drone flights is currently being assessed to explore the spatial movement patterns of the group and identify those individuals that successfully initiate changes in group movement. Alternatively, the rapid increase in vocal behavior following each ping could indicate that multiple animals exchange whistles to contact their closest social partners in the presence of an unknown stimulus. This could be expected given the role of whistles in group cohesion and coordination [2,48].

Surprisingly, in the playbacks when dolphins showed a significant increase in vocal behavior in the 5 s following the first ping (6 of 9 CEEs, S3 Fig), the severity of their acoustic response did not abate over the course of the exposure (S2 Fig). This suggests that dolphins did not habituate to successive pings (*i.e.*, show a progressive decrease in the amplitude of a vocal behavioral response after repeated exposure). This observation is surprising given that the dolphins tested in this study likely live a large portion of their lives in areas regularly ensonified by Navy sonar. Continued work is needed to confirm these initial findings; for example, future studies could compare animals' responsiveness in the Southern California Bight to nearby populations in areas where Navy sonar is not regularly present (*e.g.*, Monterey Bay). Considering whether dolphins may be learning to modify their vocal behavior in response to repeated noise exposure may provide foundational evidence to support using vocal rates as a measure of sensitization or habituation to anthropogenic stimuli, as has been done in terrestrial species [as in 49,50].

Future management decisions mitigating the impact of sonar on oceanic delphinids should consider our reported results of clear responses during CEEs when analyses were conducted at the appropriate temporal resolution. Concerning the most recent methodology for assessing the relative response severity for free-ranging marine mammals to acoustic disturbance [51]–had the acoustic response of animals to MFAS been pooled across the 10-min time window–common dolphins likely would have been assigned a behavioral response severity score of 0 (no response detected). However, when evaluating vocal behavior across a shorter time window, this animal's response would be elevated to a category three severity, which includes an increase in possible contact or alarm calls [51]. Ultimately, continued work with this (and other closely related) species should also consider how observed behavioral responses vary with respect to other contextual parameters, including behavioral state, group composition (*e.g.*, presence or absence of calves), seasonality, and environmental covariates. Concurrent efforts from this project aim to integrate passive acoustics with other remotely sensed datasets (*i.e.*, shore-based group tracking and aerial photogrammetry) to identify group-level behavioral changes and quantify exposure-response relationships [52]. Paired with energetic modeling methods [*e.g.*, 27,53,54], these data can link these observed short-term behavioral responses to long-term fitness outcomes in this species and inform effective mitigation strategies.

Our study has some limitations that can be addressed in future work. Given the close phylogenetic relationship between short-beaked and long-beaked common dolphins [55,56], the basic description of baseline vocal behavior (*i.e.*, whistle count data) was combined across subspecies in our analysis. However, recent work by Oswald et al. 2021 [57] discovered unique whistle contours in short-beaked common dolphins, suggesting that these distinctive acoustic signals could help facilitate recognition between these *Delphinus* subspecies. An increase in sample size for both long-beaked and short-beaked common dolphins would provide a more detailed understanding of their baseline vocal behavior and enable the exploration of whether each subspecies shows a differential or similar response in whistle-type usage to MFAS. Additionally, photo identification of individuals within these large, ephemeral groups is challenging. Consequently, whether individual dolphins around Catalina Island were exposed more than once to the experimental treatment is unknown. Future research could direct efforts toward other oceanic delphinids with smaller group sizes where photo-identification is feasible and reliable (*e.g.*, bottlenose or rough-toothed dolphins).

In any behavioral experiment, the response of the individual or group that is tested should be measured and interpreted over a time window that is informed by their natural behavior. We suggest that future work with other oceanic delphinid species explore baseline vocal rates a-priori and use information on vocal state-switching to inform the analysis time window over which behavioral responses are measured. Given these animals' susceptibility to frequent MFAS exposure in Navy operational areas, evaluating how repeated exposure influences responses is of particular interest. Future work should continue to explore critical factors likely to affect the probability of response among these large groups, including their behavioral state and proximity to the sound source and received level at the onset of exposure.

## Supporting information

**S1 Fig. PAMGuard whistle detection parameters.** Detection settings for the PAMGuard Whistle and Moan Detector.
(PDF)

**S2 Fig. Plots of raw per-second whistle detections for all included CEEs.** CEE-ID (year and number) is given in the bottom right of each page. The red dashed line indicates the start of the exposure period, with dashed grey lines indicating the timing of each ping (in an MFAS exposure) or control treatment. The solid gray vertical line indicates the start of the post-exposure period.
(PDF)

**S3 Fig. Boxplots of the change in whistle count from the 5 s before to the 5 s following each of the 24 pings for all CEEs (MFAS and controls).** Boxplot shows median, 25th, and 75th percentiles, with raw whistle count changes as open gray circles. The change following the first ping is shown as a red star.
(PDF)

**S1 File. CEE-by-CEE analysis.** The main manuscript presents methods and results for analyzing common dolphin vocal response to simulated mid-frequency sonar, pooling data from all CEEs. We acknowledge that some readers may be interested in additional information about each of the 19 CEEs (10 MFAS exposures and nine controls), so we have included additional methods and results for each CEE.
(PDF)

## Acknowledgments

Animal observations, octocopter flights over dolphins, close approaches, and CEEs were conducted under NMFS permits 19116 and 19091. Additionally, all activities reported in this study were reviewed and approved by the Institutional Animal Care and Use Committee (IACUC Protocol No: CRC-2021-AUP-06, CRC-2021-AUP-08). We thank the *M/V Magician* and Captain Carl Mayhugh for support in the field and the University of Southern California's Dornsife Wrigley Institute for Environment and Sustainability for their hospitality throughout this project. This project would not have been possible without the dedicated contribution of all members of the Tagless-BRS team. Thank you to Stacy DeRuiter for extensive advice on statistical modeling and analysis. Additionally, we appreciate the thoughtful feedback provided by Ari Friedlaender, Colleen Reichmuth, Peter Cook, and John Durban during analysis and manuscript preparation. We also thank Ryan Jones, who created the line drawings for Fig 1.

## Author Contributions

**Conceptualization:** Caroline Casey, Selene Fregosi, Julie N. Oswald, Vincent M. Janik, Fleur Visser, Brandon Southall.

**Data curation:** Caroline Casey, Selene Fregosi.

**Formal analysis:** Caroline Casey, Selene Fregosi, Fleur Visser.

**Funding acquisition:** Fleur Visser, Brandon Southall.

**Investigation:** Caroline Casey, Selene Fregosi, Brandon Southall.

**Methodology:** Caroline Casey, Selene Fregosi, Julie N. Oswald, Vincent M. Janik, Fleur Visser, Brandon Southall.

**Project administration:** Caroline Casey, Brandon Southall.

**Supervision:** Brandon Southall.

**Writing – original draft:** Caroline Casey, Selene Fregosi, Julie N. Oswald, Vincent M. Janik.

**Writing – review & editing:** Caroline Casey, Selene Fregosi, Julie N. Oswald, Vincent M. Janik, Fleur Visser, Brandon Southall.

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
