## [Decision Letter · Decision Letter 0]

2 Jan 2024

PONE-D-23-38783Common dolphin whistle response to experimental mid-frequency sonarPLOS ONE

Dear Dr. Casey,

Thank you for submitting your manuscript to PLOS ONE. After careful consideration, we feel that it has merit but does not fully meet PLOS ONE’s publication criteria as it currently stands. Therefore, we invite you to submit a revised version of the manuscript that addresses the points raised during the review process.

We look forward to receiving your revised manuscript.

Kind regards,

Vitor Hugo Rodrigues Paiva, Ph.D.

Academic Editor

PLOS ONE

Journal Requirements:

"Funding for this project was provided by the U.S. Navy’s Office of Naval Research (Award Numbers N000141713132, N0001418IP-00021, N000141712887, N000141912572). " 

"Funding for this project was provided by the U.S. Navy’s Office of Naval Research (Award Numbers N000141713132, N0001418IP-00021, N000141712887, N000141912572). "

"Funding for this project was provided by the U.S. Navy’s Office of Naval Research (Award Numbers N000141713132, N0001418IP-00021, N000141712887, N000141912572). "

Reviewers' comments:

Reviewer's Responses to Questions

**Comments to the Author**

1. Is the manuscript technically sound, and do the data support the conclusions?

Reviewer #1: Yes

Reviewer #2: Yes

Reviewer #3: Yes

2. Has the statistical analysis been performed appropriately and rigorously? 

Reviewer #1: Yes

Reviewer #2: Yes

Reviewer #3: Yes

3. Have the authors made all data underlying the findings in their manuscript fully available?

Reviewer #1: Yes

Reviewer #2: Yes

Reviewer #3: Yes

4. Is the manuscript presented in an intelligible fashion and written in standard English?

Reviewer #1: No

Reviewer #2: Yes

Reviewer #3: Yes

5. Review Comments to the Author

Reviewer #1: The information contained in the manuscript is useful. One has to wonder whether the results will stand if/when additional data are obtained and n is increased.

The information itself is straightforward, but the manuscript is not as clear as it could be. There also were some inaccuracies, errors, inconsistencies, missing words, and formatting issues. The manuscript should be reviewed/revised editorially, since PLOS ONE does not use a copy editor.

Some of the issues included—

• Short- and long-beaked common dolphins are separate subspecies, not species. Their correct scientific names must be used.

• The units for source level and received level must be correct and complete (dB re 1 µPa at 1 m and dB re 1 µPa, respectively) and should include rms, when applicable.

• 10 MFAS CEEs were denoted in Table 1, but the text in the manuscript specified 9 CEEs in numerous instances.

• The tense of nouns and verbs should be the same within a given sentence, as should the person.

• Commas should be used consistently and correctly throughout. In some instances, commas were not used when they should have been (i.e., after introductory clauses).

• Hyphens should not be used for nouns (i.e., ‘in 5 sec’ instead of ‘in 5-sec’) but should be used for adjectives (i.e., ‘in a 5-sec bin’ instead of ‘in a 5 sec bin’).

• Abbreviations should be used consistently (i.e., second, sec, and s were all used).

• “Table” should be capitalized when cited in the text.

• References should be cited in the correct order (numerically ascending), format (when the name also is used in the text), and with the correct punctuation (brackets [] instead of parentheses () for PLOS ONE).

• Capitalization for headings/headers and bolding of captions should be consistent.

• Extra spaces should be deleted within and between sentences.

Also, information regarding the IACUC review and approval was missing in various sections of the manuscript.

Please see the pdfs for specific comments and questions regarding the manuscript and supplementary docs.

Reviewer #2: Casey et al. aimed to measure the impact of Navy MFAS sonar on free-ranging dolphins in comparison to baseline variability data of acoustics. They utilize a network of acoustic buoys and controlled exposure experiments measure acoustic disturbance of free ranging short-beaked and long-beaked common dolphin whistles. CEEs were conducted off the coast of Santa Catalina Island and data were analyzed for vocal state changes. The authors found a significant vocal state change in the 5 seconds post CEEs. Interestingly, they give information on potential habituation of the animals as well.

Overall, it is a well written paper looking at vocal state changes as a tool for measuring disturbance. There are a few errors and areas that transition phrases are redundant. There are some areas where the authors have accidentally put in an extra space between words.

Line 19: Considering we do not have data that states that millions hear and are affected by Navy MFAS in particular, I would suggest rephrasing this first sentence or adding a citation.

Line 222-223: While the author’s state that the distance between the drone and the octocopter were estimated every thirty minutes, it would be beneficial for the authors to state how high the drone was flying on average.

Line 427: Consider deleting “however” after Note.

Line 448: consider replacing en dash with an em dash.

Line 449: 10 MFAS CEEs were conducted but only 2 in Dolphinus delphis, why? I would suggest at least one line of explanation on this.

Line 483: cee is not capitalized in this line but is in front of the word type in the line before. I would suggest making this congruent in the paragraph.

Line 618: There is an extra space between the last word of the sentence and the period.

Line 647: There is an extra space between the words “sizes” and “where”.

Reviewer #3: In this manuscript, the authors investigate the impact of mid-frequency active sonar (MFAS) on the whistle production of common dolphins in Southern California. The study uses a network of drifting acoustic buoys in a controlled exposure experiment to analyze the dolphins' acoustic reactions to sonar exposure over various time frames, to understand both longer-term and immediate changes in sound production.

The manuscript is detailed and informative but can be challenging to follow due to its complexity. Simplifying the language or reorganizing the data presentation could enhance readability. The authors do a good job arguing that the main objective is to clearly understand variability in baseline and examine differences in whistle count compared to baseline. However, some aspects are still unclear and could help the readability and better understanding of the scale.

To better grasp the scope of the study, it would be helpful to know: How much total recording hours were collected? How much was actually used in the analysis? Was the use of WMD validated with manually labeled data?

Although automated tools have gotten better, they still often bring a large amount of errors compared to manual labeling which is still recommended depending on the task. The validity of the results would be improved if the authors could use a subset of their data for manual analysis as a reliability measure of the software approach.

The abstract effectively summarizes a lot of information but could be enhanced by a clearer statement of the main findings beyond “exhibited an acute and dramatic change in acoustic behavior in the 5-seconds following exposure to experimental” Is the metrics only a count of whistle per overlapping windows?

L.157 - To what extent are the animals already accustomed to the signals. For how long? How many generations? What differences in dispersion, sample rate, etc, between actual MFAS pings and experimental conditions could affect the results?

L.180 - Regarding the use of drones for the experiment, specific details such as the type of drones, their number, operational height, and potential disturbances (visual or sonic) at the water level would be valuable.

L.180-190 - Good methodology observation

L.227 - More clarification needed

L.260 - Good methodological point

L.270: Why use different recording hardware? Any baseline to address differences in recordings?

L.282: More clarification needed

Fig 1: It would be beneficial to add scale information to the figure.

L.577 - The lack of elevation in whistle count during the 1.6s signal broadcast is intriguing. Further explanations on this observation and its implications would be insightful. Was the sonar frequency removed from the recorded audio? Extend on the possible use of full duplex vs single duplex for future experiments.

L.590 - Interesting hypothesis, how could this be tested in the future?

L.607 - This point warrants earlier discussion in the manuscript for better context.

L.618 - Extra space before comma.

How do the two species compare in terms of vocal reactions?

Overall, the manuscript is well-structured and addresses an important topic. The methodology is very thorough but some details should be clarified. By addressing these points, the study could significantly enhance its contribution to understanding dolphin behavior under anthropogenic disturbance. The clarity of language and presentation is good. Ensuring the results and discussion sections are as robust and clear as the earlier sections will be crucial for the manuscript's overall strength.

6. PLOS authors have the option to publish the peer review history of their article (what does this mean?). If published, this will include your full peer review and any attached files.

Reviewer #1: No

Reviewer #2: No

Reviewer #3: No

---

## [Author Response · Author response to Decision Letter 0]

26 Feb 2024

Dear Vitor Hugo Rodrigues Paiva and the editorial team,

We greatly appreciate your response to our submitted manuscript (Common dolphin whistle response to experimental mid-frequency sonar – PONE-D-23-38783). It was helpful and gratifying to receive reviewer comments on our study that were so thoughtful and meticulous. We appreciate that the editorial team believes that this is a unique paper that should be of broad interest to the readers of PLOS One. 

We have spent considerable time reflecting on and revising this manuscript based on the detailed recommendations of each reviewer. These changes can be seen in the “Revised Manuscript with Track Changes” document.

With respect to the specific Journal Requirements, we have made the following requested changes:

The Title, Author, and affiliations have been updated. Additionally, we have carefully gone through PLOS ONE’s style requirements, and have made several changes to the manuscript.

"Funding for this project was provided by the U.S. Navy’s Office of Naval Research (Award Numbers N000141713132, N0001418IP-00021, N000141712887, N000141912572). " 

This statement is correct. The funders had no role in study design, data collection and analysis, decision to publish, or preparation of the manuscript. We have removed the funding information from the Acknowledgements section and can re-include this in our cover letter. 

"Funding for this project was provided by the U.S. Navy’s Office of Naval Research (Award Numbers N000141713132, N0001418IP-00021, N000141712887, N000141912572). "

"Funding for this project was provided by the U.S. Navy’s Office of Naval Research (Award Numbers N000141713132, N0001418IP-00021, N000141712887, N000141912572). "

We apologize for the confusion. We have removed the funding statement from the main body of the manuscript. The funding statement that you have here is correct. 

We have redesigned the figure based on the map provided by NASA Earth Observatory (public domain) that complies with the CC BY 4.0 license. The updated Figure 1. is now reflected in the manuscript. We have also added the following to the figure caption. “The map was inspired by images obtained from the NASA Earth Observatory (public domain), is not drawn to scale, and is for illustrative purposes only”. 

This has been done. 

We have carefully gone through our references and have updated the formatting. We do not believe we have any retracted references in our reference list.

Additionally, all the reviewers made minor suggestions to strengthen the manuscript, and we have addressed each in turn. Our specific responses to reviewers are provided below. Given these changes, we hope that PLOS One finds our manuscript suitable for publication. Please do not hesitate to contact me if you require any additional information.

Specific comments to reviewers can be found below:

Reviewer #1: The information contained in the manuscript is useful. One has to wonder whether the results will stand if/when additional data are obtained and n is increased.

We certainly aim to address this in the future in our ongoing BRS efforts and appreciate the supportive feedback. 

The information itself is straightforward, but the manuscript is not as clear as it could be. There also were some inaccuracies, errors, inconsistencies, missing words, and formatting issues. The manuscript should be reviewed/revised editorially, since PLOS ONE does not use a copy editor.

Some of the issues included—

• Short- and long-beaked common dolphins are separate subspecies, not species. Their correct scientific names must be used.

Thank you for bringing this to our attention. This has been corrected throughout the manuscript based on the Marine Mammal Society Taxonomic list of marine mammal species subspecies. 

• The units for source level and received level must be correct and complete (dB re 1 µPa at 1 m and dB re 1 µPa, respectively) and should include rms, when applicable

Thank you, this has been adjusted throughout the manuscript where appropriate.

• 10 MFAS CEEs were denoted in Table 1, but the text in the manuscript specified 9 CEEs in numerous instances.

We apologize for the confusion on this. We realize that there were some details here that needed to be resolved. Specifically, we had to exclude one of the CEEs (2021_11) from our modeling work because of the low number of whistles detected during this experiment. The following information has been added to the text of the manuscript for clarity: “For the changepoint analysis, all 10 MFAS experiments were included. However, for the assessment of changes in whistle behaviors across different time scales, CEE 2021_11 (conducted with long-beaked common dolphins) had to be excluded because the overall lack of detected whistles could not be successfully modeled. Consequently, the modeling results include nine controls and nine MFAS experiments. ”

• The tense of nouns and verbs should be the same within a given sentence, as should the person.

• Commas should be used consistently and correctly throughout. In some instances, commas were not used when they should have been (i.e., after introductory clauses).

• Hyphens should not be used for nouns (i.e., ‘in 5 sec’ instead of ‘in 5-sec’) but should be used for adjectives (i.e., ‘in a 5-sec bin’ instead of ‘in a 5 sec bin’).

• Abbreviations should be used consistently (i.e., second, sec, and s were all used).

• “Table” should be capitalized when cited in the text.

• References should be cited in the correct order (numerically ascending), format (when the name also is used in the text), and with the correct punctuation (brackets [] instead of parentheses () for PLOS ONE).

• Capitalization for headings/headers and bolding of captions should be consistent.

• Extra spaces should be deleted within and between sentences.

We thank the reviewer for these detailed comments. We have carefully gone through the manuscript and have addressed each of their edits. These changes can be seen within the “Revised Manuscript with Track Changes” document. 

Also, information regarding the IACUC review and approval was missing in various sections of the manuscript.

We apologize for this oversight. Our IACUC information has been added to the Acknowledgments section. The text now reads: “Additionally, all activities reported in this study were reviewed and approved by the Institutional Animal Care and Use Committee (IACUC Protocol No: CRC-2021-AUP-06, CRC-2021-AUP-08).”

Please see the pdfs for specific comments and questions regarding the manuscript and supplementary docs.

Many suggestions were made directly in an associated PDF (Reviewer #1). Those changes (unless noted otherwise below) were made to the revised version of the manuscript and greatly improved the clarity. We thank Reviewer 1 for the time they spent reviewing our work. 

- With respect to their comments about the consistency of the y-axis in Fig. 4b and Fig 4c, we decided to leave the figure as is, as the intention of the figure is to highlight the relative changes between the pre-exposure and exposure period for each individual plot, rather than compare across the plots. Keeping the axes as they are allows maximum resolution for across period comparisons.

- We will upload the data to NCEI as well since the project was funded by ONR.

- With respect to their comment about the validity of pooling the two subspecies acoustic data for baseline analysis, we decided not to pursue this approach since they regularly occur in mixed groups, and our sample size of Delphinus delphis delphis was relatively small. Further work looking at subspecies-specific differences in whistle production is still needed but is not within the scope of this paper. 

Reviewer #2: Casey et al. aimed to measure the impact of Navy MFAS sonar on free-ranging dolphins in comparison to baseline variability data of acoustics. They utilize a network of acoustic buoys and controlled exposure experiments measure acoustic disturbance of free ranging short-beaked and long-beaked common dolphin whistles. CEEs were conducted off the coast of Santa Catalina Island and data were analyzed for vocal state changes. The authors found a significant vocal state change in the 5 seconds post CEEs. Interestingly, they give information on potential habituation of the animals as well.

Overall, it is a well written paper looking at vocal state changes as a tool for measuring disturbance. There are a few errors and areas that transition phrases are redundant. There are some areas where the authors have accidentally put in an extra space between words.

Thank you for this positive review of our manuscript. To address reviewers 1 and 2’s comments, we have gone through the manuscript carefully to correct any editorial errors. We hope that this makes things easier to follow and clearer for the reader.

Line 19: Considering we do not have data that states that millions hear and are affected by Navy MFAS in particular, I would suggest rephrasing this first sentence or adding a citation.

We agree and have changed this sentence to be more general.

Line 222-223: While the author’s state that the distance between the drone and the octocopter were estimated every thirty minutes, it would be beneficial for the authors to state how high the drone was flying on average.

Thank you for this suggestion. We have added the following details to the text to address both reviewer 2 and 3’s point: “The animals’ location was known from an associated octocopter drone flight (APO-42, Aerial Imaging Solutions) that utilized a micro 4/3 digital camera (Olympus E-PM2) and 25 mm lens (Olympus M. Zuiko F1.8) mounted to a gimbal. The octocopter flew at approximately 60 m directly above the dolphins to provide sufficient pixel resolution while also decreasing the potential for disturbance (see [34] for details).”

Line 427: Consider deleting “however” after Note.

We appreciate this suggestion and we have made this change.

Line 448: consider replacing en dash with an em dash.

We appreciate this suggestion and we have made this change.

Line 449: 10 MFAS CEEs were conducted but only 2 in Dolphinus delphis, why? I would suggest at least one line of explanation on this.

The reviewer here caught an important mistake in our text. While Table 1. reflected the correct proportion of CEEs for both sub-species (7 of 10 with D. d. bairdii and 3 of 10 with D. d. delphis), the text did not. The text

---

## [Decision Letter · Decision Letter 1]

17 Mar 2024

PONE-D-23-38783R1Common dolphin whistle responses to experimental mid-frequency sonarPLOS ONE

Dear Dr. Casey,

Thank you for submitting your manuscript to PLOS ONE. After careful consideration, we feel that it has merit but does not fully meet PLOS ONE’s publication criteria as it currently stands. Therefore, we invite you to submit a revised version of the manuscript that addresses the points raised during the review process.

We look forward to receiving your revised manuscript.

Kind regards,

Vitor Hugo Rodrigues Paiva, Ph.D.

Academic Editor

PLOS ONE

Journal Requirements:

Reviewers' comments:

Reviewer's Responses to Questions

**Comments to the Author**

1. If the authors have adequately addressed your comments raised in a previous round of review and you feel that this manuscript is now acceptable for publication, you may indicate that here to bypass the “Comments to the Author” section, enter your conflict of interest statement in the “Confidential to Editor” section, and submit your "Accept" recommendation.

Reviewer #1: (No Response)

Reviewer #2: All comments have been addressed

2. Is the manuscript technically sound, and do the data support the conclusions?

Reviewer #1: Yes

Reviewer #2: Yes

3. Has the statistical analysis been performed appropriately and rigorously? 

Reviewer #1: Yes

Reviewer #2: Yes

4. Have the authors made all data underlying the findings in their manuscript fully available?

Reviewer #1: Yes

Reviewer #2: Yes

5. Is the manuscript presented in an intelligible fashion and written in standard English?

Reviewer #1: Yes

Reviewer #2: Yes

6. Review Comments to the Author

Reviewer #1: There are a few outstanding issues with noun/verb tense, extra spaces, not using a space between 1 and µPa, and inconsistent formats. There also were a few places where the wording could have been a bit clearer in the revised text. All of these are minor in nature and not time consuming to fix. Given that PLOS ONE does not use a copy editor, these minor revisions would be left to the authors to address. Please see the comments in the attached pdf.

Reviewer #2: (No Response)

7. PLOS authors have the option to publish the peer review history of their article (what does this mean?). If published, this will include your full peer review and any attached files.

Reviewer #1: No

Reviewer #2: No

---

## [Author Response · Author response to Decision Letter 1]

21 Mar 2024

Line 21: Reviewer’s comment: “Do you mean “that occur in and around Navy operating areas?”

Response: Yes, we have changed the wording slightly here to make this clearer. It now reads, “Oceanic delphinids that occur in and around Navy operational areas…”

Line 50: Reviewer’s comment: Remove the word “activity.”

Response: This has been done.

Line 58: Reviewer comment: Change the order of wording in the sentence.

Response: This has been done; the sentence now reads, “Among cetaceans, oceanic delphinids represent an essential and logistically challenging group to evaluate how anthropogenic noise impacts their vocal behavior.”

Line 61: Reviewer comment: Delete extra space.

Response: This has been done. 

Line 76: Reviewer comment: Replace the word “have” with “has.”

Response: This has been done.

Line 93: Reviewer comment: Change the order of words in this sentence.

Response: This has been done. The sentence now reads, “Previous studies on acoustic responses of oceanic dolphins to Navy sonar have observed shifts in frequency-specific components of whistle contours, increasing or decreasing calling rate, increasing call amplitude, and even mimicry of MFAS elements.”

Line 102: Reviewer comment: sp. Shouldn’t be italicized. 

Response: This has been changed. 

Line 167: Reviewer comment: 1 �Pa/V should be revised to include a space throughout the ms.

Response: This has been addressed throughout the ms, including Table 1. 

Line 243: Reviewer comment: change to 1.6-km threshold.

Response: This has been done.

Line 454: Reviewer comment: change the wording of the sentence.

Response: This has been done; the sentence now reads, “However, for the assessment of changes in whistle behavior across different time scales, CEE 2021_11 (conducted with long-beaked common dolphins) had to be excluded because the limited number of detected whistles could not be successfully modeled.”

Line 516: Reviewer comment: I believe, based on the caption format for PLOS ONE, this should be unbolded (similar to B), and a bolded general title of the figure should be added. 

Response: This has been done, the figure caption now reads “Fig 5. Acoustic response of common dolphins to the first ping of experimental sonar. (A) Spectrogram example of 5 s before and 5 s after the first ping for MFAS CEE 2021_08 illustrates the significant increase in whistle count immediately following the cessation of the ping. The focal group was comprised of approximately 30 long-beaked common dolphins. The MFAS signal can be seen between 3 and 4 kHz. (B) Boxplot of the change in whistle count from the 5 s before to the 5 s following each of the 24 pings for CEE 2021_08. Boxplot shows median, 25th, and 75th percentiles, with raw whistle count changes as open gray circles. The change following the first ping is shown as a red star. “

Line 601: Reviewer comment: change to 5 s based on the format of the rest of the manuscript.

Response: This has been done.

Line 635: Reviewer comment: The date is missing, and the 57 should be placed after the date rather than at the end of the sentence.

Response: This has been changed. The sentence now reads “However, recent work by Oswald et al. 2021 [57] discovered unique whistle contours in short-beaked common dolphins, suggesting that these distinctive acoustic signals could help facilitate recognition between these Delphinus subspecies.”

---

## [Editor Report · Decision Letter 2]

28 Mar 2024

Common dolphin whistle responses to experimental mid-frequency sonar

PONE-D-23-38783R2

Dear Dr. Casey,

We’re pleased to inform you that your manuscript has been judged scientifically suitable for publication and will be formally accepted for publication once it meets all outstanding technical requirements.

Kind regards,

Vitor Hugo Rodrigues Paiva, Ph.D.

Academic Editor

PLOS ONE